# Understanding and predicting COVID-19 clinical trial completion *vs.* cessation

**Magdalyn E. Elkin<sup>☯</sup>, Xingquan Zhu<sup>☯</sup>***

Department of Computer & Electrical Engineering and Computer Science, Florida Atlantic University, Boca Raton, FL, United States of America

☯ These authors contributed equally to this work.
* xzhu3@fau.edu

## Abstract

As of March 30 2021, over 5,193 COVID-19 clinical trials have been registered through Clinicaltrial.gov. Among them, 191 trials were terminated, suspended, or withdrawn (indicating the cessation of the study). On the other hand, 909 trials have been completed (indicating the completion of the study). In this study, we propose to study underlying factors of COVID-19 trial completion *vs.* cessation, and design predictive models to accurately predict whether a COVID-19 trial may complete or cease in the future. We collect 4,441 COVID-19 trials from ClinicalTrial.gov to build a testbed, and design four types of features to characterize clinical trial administration, eligibility, study information, criteria, drug types, study keywords, as well as embedding features commonly used in the state-of-the-art machine learning. Our study shows that drug features and study keywords are most informative features, but all four types of features are essential for accurate trial prediction. By using predictive models, our approach achieves more than 0.87 AUC (Area Under the Curve) score and 0.81 balanced accuracy to correctly predict COVID-19 clinical trial completion *vs.* cessation. Our research shows that computational methods can deliver effective features to understand difference between completed *vs.* ceased COVID-19 trials. In addition, such models can also predict COVID-19 trial status with satisfactory accuracy, and help stakeholders better plan trials and minimize costs.

## Introduction

Since its outbreak in late 2019, Coronavirus Disease 2019 (COVID-19) has spread worldwide, and has become a major global pandemic. As of February 2021, there have been over 100 million cases, with a total of over two million deaths worldwide [1]. In order to win the battle against COVID-19, numerous research have been taken in developing vaccines, drugs, re-purpose of existing drugs, and devices [2]. Among all these studies, clinical trials are one of the unavoidable steps to determine the validity of any intervention, treatment, or test on human subjects.

As of March 2021, the United States Food and Drug Administration (FDA) has approved one intervention, Remdesivir, to treat COVID-19 [3]. On the other hand, in December 2020,

**Data Availability Statement:** The data used in this study are available on GitHub at https://github.com/maggieelkin/ClinicalTrialReports.

**Funding:** XZ, National Science Foundation, IIS-2027339, https://www.nsf.gov/awardsearch/showAward?AWD_ID=2027339 XZ, National

Science Foundation, IIS-1763452, https://www.nsf.
gov/awardsearch/showAward?AWD_ID=1763452
The funders had no role in study design, data
collection and analysis, decision to publish, or
preparation of the manuscript.

**Competing interests:** The authors have declared
that no competing interests exist.

the FDA created Emergency Use Authorization (EUA) for two vaccines. Within the first
month of the U.S. COVID-19 vaccination program, approximately 4% of the total population
has been vaccinated [4]. For a vaccine or treatment to receive an EUA or FDA approval, ran-
domized controlled clinical trials are necessary to provide evidence of safety and efficacy.
Often multiple large scale trials are required. Three separate randomized control clinical trials
provided data to the FDA to approve Remdesivir [3]. A single large randomized clinical trial
with 30,000 participants provided body of evidence for the Moderna COVID-19 vaccine EUA
[5]. Similarly, a randomized clinical trial with 43,000 participants provided the body of evi-
dence for the Pfizer COVID-19 vaccine EUA [6].

Since 2007, the FDA Amendments Act (FDAAA) requires clinical trials to be registered to
the online database, ClinicalTrials.gov, if a trial has one or more sites in the U.S., involves an
FDA investigational new drug/device, or involves a drug/device product manufactured in the
U.S. and exported for research. As of March 2021, there are 371,776 registered trials with
about 30,000 trials being registered every year since 2018 [7]. Unfortunately, not all trials are
completed and achieve the intended goals. Researchers, in previous studies, have found that
10–12% of clinical trials are terminated [8–10] due to numerous reasons, such as insufficient
enrollment, scientific data from the trial, safety/efficacy concerns, administrative reasons,
external information from a different study, lack of funding/resources, and business/sponsor
decisions, etc. [8, 9, 11].

COVID-19 clinical trials are necessary, not only to provide safety and efficacy data for treat-
ments and vaccines, but also to understand the disease. As a major global epidemic, it becomes
imperative to understand what type of trials are likely going to be successful. Ceased clinical
trials (including terminated, suspended, and withdrawn trials) are costly and often represent a
loss of resources, including financial costs, drugs, administrative efforts, and time. As future
outbreaks of COVID-19 are likely even after the current pandemic has declined, it is critical to
optimise efficient research efforts [12]. Machine learning and computational approaches [13]
have been developed for COVID-19 healthcare applications, and deep learning techniques
have been applied to medical imaging processing in order to predict outbreak, track virus
spread and for COVID-19 diagnosis and treatment [14]. It would be helpful to design compu-
tational approaches to predict whether a COVID-19 clinical trial may complete or not, such
that stakeholders can leverage the predictions to plan resources, reduce costs, and minimize
the time of the clinical study.

In this study, our main objective is to create features to model COVID-19 clinical trial
reports. Using feature engineering and feature selection techniques, our research will deter-
mine features associated to trial completion *vs.* cessation. Using machine learning algorithms
and ensemble learning, we can predict completion or cessation of a COVID-19 clinical trial.
These computational approaches allow for a deeper understanding and prediction of factors
associated to successful COVID-19 clinical trials.

## Related work

Previous research has studied the usage of features from structured fields and unstructured
fields in order to understand factors associated to clinical trial termination. Using manual
examination and categorization of terminated trials, research has been conducted by analysis
the "Why Study Stopped" field in clinical trial reports. These studies provide statistical data
based on the number of trials terminated due to primary reasons, such as insufficient enroll-
ment [8, 9]. Unfortunately, there is little detailed information on the factors that lead to termi-
nation, such as specific characteristics of the trial leading to termination. Moreover, a previous
study also reported that as much as 10% of terminated trials do not utilize the "Why Study

Stopped" field [9], leaving structured data based approaches inapplicable to analyze those reports. Using machine learning to model clinical trial terminations allows for a greater understanding of the specific factors that may lead to terminated clinical trials. These models can also be applied to current or planned trials to understand their probability of completion *vs* termination. Previous termination prediction studies [15, 16], along with our previous research [10] demonstrate the predictive power of structured and unstructured variables in order to predict if a clinical trial is likely to terminate.

A separate study modeled clinical trial terminations by investigating the drug interventions associated with trials that failed for safety concerns *vs.* FDA approved drugs. Using target based and chemical based features, they created a model that can predict if a drug is likely to fail for toxicity concerns [17]. While this type of model can accurately predict interventions that fail for toxicity concerns, many trials stop prematurely due to other concerns. Additionally, trials may terminate due to efficacy concerns of the medical intervention, which wouldn't be captured when modeling drug toxicity of the intervention alone.

These previous studies are all aimed at generic clinical trials, but not specialized in a specific domain. One of the main goals of our study is to model clinical trials *w.r.t.* COVID-19. The domain specific modeling allows us to clearly represent unique components and challenges faced by COVID-19 clinical research, such as emerging COVID-19 information, intervention assessments, and interpretations of symptom resolution [18].

In our previous research [10], we designed a predictive model to predict completed *vs.* terminated trials using a large testbed collected from the entire clinicaltrials.gov database. In this study, we propose to design new features to characterize COVID-19 trials, and understand factors associated to clinical completion *vs.* cessation. Because COVID-19 is a relatively novel disease, very few trials have been formally terminated, we consider three types of trials, including terminated, withdrawn, and suspended, as cessation trials. These trials represent research efforts that have been stopped/halted for particular reasons and represent research efforts and resources that weren't successful.

Since the majority of COVID-19 trials study medical interventions, we introduce drug features to specify the medical interventions under trail instigation. The choice of intervention within a trial has a high consequence, not only for toxicity reasons, but also for efficacy reasons. Many COVID-19 interventions under study are for drug re-purposing, and any trial studying an intervention that is deemed in-effective may ultimately result in a premature stop.

## Contribution

Using drug features combined with other types of features, including statistics features, keyword features and embedding features, 693 features are created to represent each clinical trial. Using ReliefF feature selection, we will study features associated to the trial completion *vs.* cessation, and provide interpretable knowledge for domain experts to understand what types of trials are more/less likely to complete.

In addition to the understanding of COVID-19 clinical trials using features, our research also proposes to predict trial completion *vs.* cessation, by using sampling and ensemble learning. We employ different sampling ratios to change data distributions, and use ensemble of multiple classifiers to predict the probability of completion *vs.* cessation of a clinical trial. The experiments confirm that our model achieves over 0.87 AUC scores and over 0.81 balanced accuracy for prediction.

Table 1 summarizes existing methods for clinical trial termination study and outlines their difference. Comparing to existing research in the field, the main contribution of our study is as follows.

**Table 1. A summary and comparison between different methods (including proposed research) for clinical trial study.**

| Method & Paper | Domains | Drug Features | Keyword Features | Embedding Features | Elig. & Des. Features[†] | Admin. & Sty. Features[‡] | Data Source | Objective |
|---|---|---|---|---|---|---|---|---|
| Manual examination & categorization [8, 9] | Generic | | | | | ✓ [8] ✓◊ [9] | ClinicalTrials.gov MEDLINE [8] | Trial termination attribution |
| Machine learning & RandomForest [17] | Generic | ✓* | | | | | ClinicalTrials.gov DrugBank | Drug Toxicity |
| Text mining & RandomForest [15, 16] | Generic | | ✓ [15] ✓** [16] | | | ✓ | ClinicalTrials.gov | Trial Terminations |
| Ensemble learning & Four classifiers [10] | Generic | | ✓ | ✓ | ✓ | ✓ | ClinicalTrials.gov | Trial Characteristics & Terminations |
| Proposed research Ensemble learning | COVID-19 | ✓ | ✓ | ✓ | ✓ | ✓ | ClinicalTrials.gov | Trial Completion *vs* & Cessation |

[†] Eligibility features and study design features.

[‡] Administrative features and study information features.

◊ Create "Ontology of termination" using administrative features and study information features.

* Drug features are based on drug chemical compound, drug gene target, and gene-gene interaction.

** Keyword features are latent topics derived from Detailed Description field, using Latent Dirichlet Allocation (LDA).

- COVID-19 clinical trial benchmark: Our research delivers a COVID-19 benchmark for clinical trial completion *vs.* cessation study. The benchmark, including features and supporting documents, are published online for public access.

- Clinical trial features: Our research proposes the most extensive set of features for clinical trial reports, including features to model trial administration, study information and design, eligibility, keywords, and drugs *etc.* In addition, our research also uses embedding features to model unstructured free texts in clinical trial reports for prediction.

- Predictive modeling of COVID-19 trials: Our research is the first effort to model COVID-19 clinical trial completion *vs.* cessation. By using ensemble learning and sampling, our model achieves over 0.87 AUC scores and over 0.81 balanced accuracy for prediction, indicating high efficacy of using computational methods for COVID-19 clinical trial prediction.

## Materials and methods

### COVID-19 clinical trials and inclusion criteria

A total of 4,441 COVID-19 clinical trials were downloaded, in XML format, from ClinicalTrials.gov in January, 2021. These trials are retrieved from the online database by using search keywords "COVID-19", "SARS-CoV-2" or "COVID".

Clinical trials can have sites in multiple countries. For our analysis, the main country for each trial, *i.e.* the one with the maximum number of sites, is recorded. In case of a tie, the first country listed is recorded. The entire dataset covers 108 different countries/regions, and the top 10 countries are reported in Table 2.

Trial status field represents the overall recruitment of the study. The recruitment status could be one of: "Not yet recruiting", "Recruiting", "Enrolling by invitation", "Active", "not recruiting", "Completed", "Suspended", "Terminated", and "Withdrawn". If a clinical trial has expanded access, the status could also be one of: "Approved for Marketing", "No Longer Available", or "Available" [19].

**Table 2. The top 10 countries/regions with the most registered COVID-19 clinical trials.**

| Countries/Regions | # Trials | Countries/Regions | # Trials |
|---|---|---|---|
| United States | 901 | China | 154 |
| France | 571 | Egypt | 153 |
| Italy | 185 | Canada | 147 |
| United Kingdom | 182 | Turkey | 147 |
| Spain | 155 | Brazil | 123 |

For all COVID-19 clinical trials, the distribution of their status is reported in Fig 1. Because COVID-19 is a relatively novel disease, majority trials (50.24%) are still recruiting, and only 14.14% of trials have completed. Completed trials are those that have concluded normally and participants are no longer being examined. Terminated trials are those that had participants enrolled, yet stopped prematurely. Suspended trials are the ones that halted prematurely, but may resume in the future. Withdrawn studies are those that stopped prematurely, prior to enrollment of any participants. These three statuses, (Terminated, Suspended and Withdrawn), represent unintended cessation of clinical trials. In total, cessation trials represent 3.24% of all COVID-19 clinical trials registered on the ClinicalTrials.gov.

Fig 2 reports the data inclusion criteria. The trials included in analysis are those whose status are marked as "Completed" (completion trials) and those that are marked as "Terminated", "Withdrawn", or "Suspended" (cessation trials). In total, the final dataset has 772 clinical trials with 81.34% completion trials and 18.65% cessation trials.

## COVID-19 clinical trial features

**Statistics features.** Statistics features model clinical trials with respect to administrative, study information, study design, and eligibility criteria, which are extracted from the clinical trial reports by taking different part of information into consideration [10].

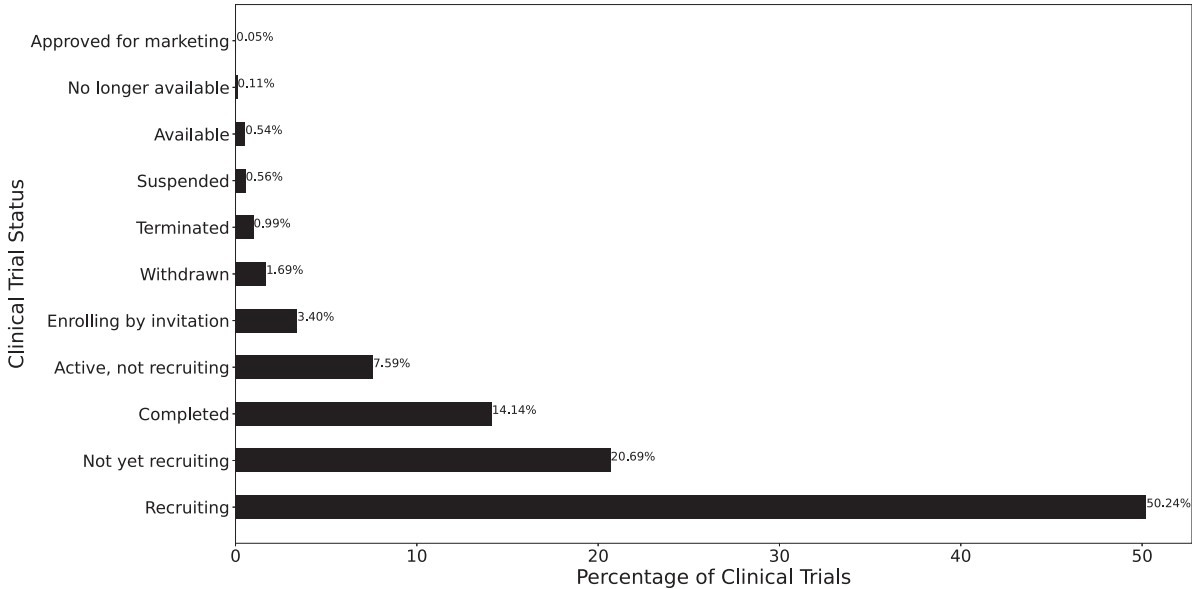

**Fig 1. Status field distributions of all COVID-19 clinical trials.**

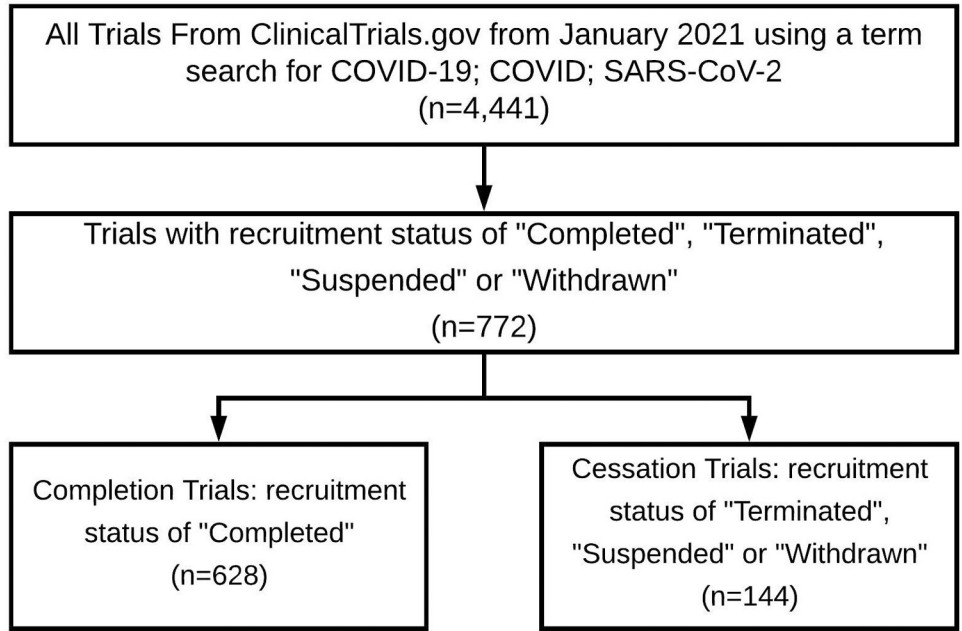

**Fig 2. Data inclusion criteria for COVID-19 clinical trials.**

*Administrative features.* Administrative features extract information about the clinical trial administration, which may directly impact on a trial, depending on the funding source and the person responsible for the clinical trial's management. Administrative features include industry sponsorship, industry collaborator sponsorship, the number of collaborators, number of officials and the type of responsible party. The sponsor of the trial is the main organization responsible for overseeing the clinical trial. A collaborator is an organization that may provide support to the trial. Collaboration can be key for a trial completion, as collaboration between entities can increase funds and resources required to reach the target enrollment [20]. Sponsors and/or collaborators can be industry or non-industry. Non-industry organizations include government organizations such as National Institutes of Health (NIH). Officials are those responsible for the scientific conduct of the trial's protocol [19]. The responsible party field indicates if the sponsor or principal investigator is the main responsible party for the trial.

Figs 3 and 4 report the distribution of trials with respect to industry sponsors and industry collaborators, respectively. The statistics show that, on average, industry sponsors/collaborators have a higher percentage in cessation trials. This means that trials involving industry sponsorship or collaborators are more likely to be ceased, rather than be completed.

*Study information features.* Study information features intend to capture basic clinical trial information, such as whether the trial has expanded access, Data Monitoring Committee (DMC) regulation, FDA regulation, the phase of the trial, if the study is interventional or observational, and if the trial's main country is USA. Trials with expanded access are those studying an investigational drug for patients not qualify for enrollment in a clinical trial [19]. Expanded access participants are typically those having serious conditions and exhausted all other medical interventions. Data monitoring committees are independent scientists overseeing the data collected in a trial to monitor health and safety of participants [19]. In the case that the intervention has shown to be not safe or effective, a DMC will recommend to terminate the trial. FDA regulation indicates that the trial includes an intervention under FDA regulation, such as investigational new drug trials. DMC regulation and/or FDA regulation may

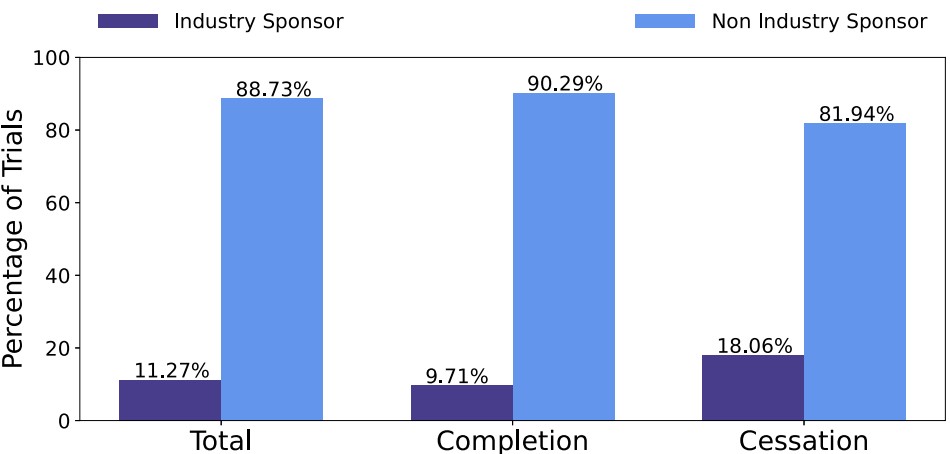

**Fig 3. Distribution of trials that have industry sponsor with respect to all trials, completion trials, and cessation trials, respectively.**

pose strict safety and efficacy guidelines for participants within the study. Trials do not meet these guidelines under FDA and/or DMC regulation will halt prematurely.

The study phase of a trial indicates different stages of clinical research. In the case of device or behavioral interventions, and observation trials, the trial could have no phase. For medical intervention trials, the phase of a trial varies from 1 to 4. Phase 1 trials are initial studies aiming at determining the initial dosage pharmacologic action and side effects with increasing doses. Phase 1 trials can include healthy participants. Phase 2 trials evaluate the effectiveness of the intervention with regards to participants with a particular condition under study. Phase 3 trials are longer trials aiming to evaluate longer term safety and efficacy of the intervention. Phase 4 trials are those for FDA-approved drugs for additional long term data on safety and efficacy [19].

Interventional clinical trials are those that introduce a behavioral, device or medical intervention to participants. For COVID-19 clinical trials, these are often medical interventions aiming at treating COVID-19. Observational trials are those not introducing an intervention, and

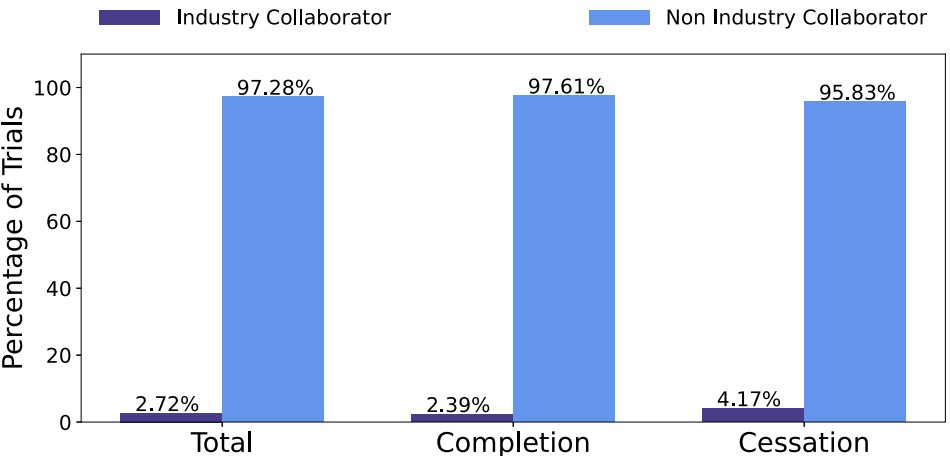

**Fig 4. Distribution of trials that have industry collaborator with respect to all trials, completion trials, and cessation trials, respectively.**

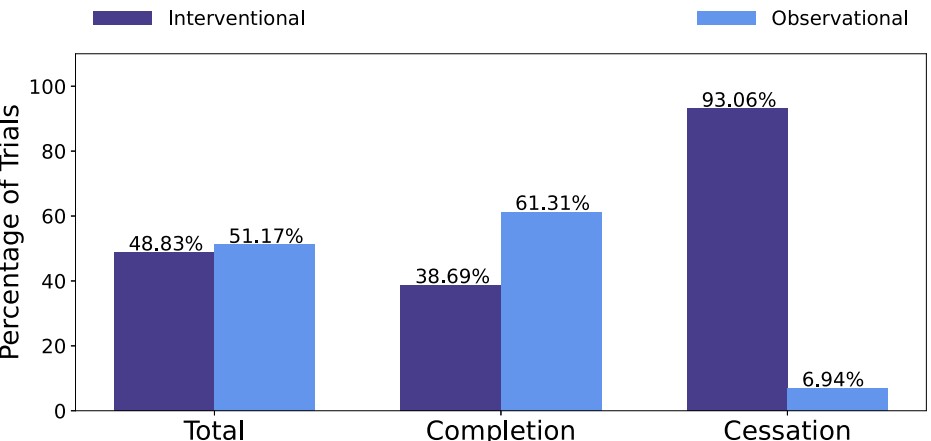

**Fig 5. Distribution of trials that are interventional or observational with respect to all trials, successful trials and unsuccessful trials.**

participants are observed to gather data and information. The distributions of interventional to observational trials for all trials, completion trials, and cessation trials are reported in Fig 5.

Fig 5 shows that, overall, 48.83% COVID-19 trials are interventional, whereas for cessation trials, 93.06% are interventional. Interventional trials have more obstacles to hurdle, such as proving their intervention is safe and effective and completing their participant enrollment requirements.

Clinical trials can have sites in multiple countries. For trials in our dataset, the most common country listed as a trial's site is recorded as the main country. In the case of a tie, the first country listed as a site is recorded. For trials in the final dataset, there are 62 unique countries. A feature is created to capture if the trial's main country is USA or outside USA. A total of 14.12% of trials in the dataset are in USA. Among all cessation trials, 27.08% of them are in USA. The distributions of outside USA to inside USA for all trials, completion trials, and cessation trials are reported in Fig 6.

*Study design features*. Study design features extract information about whether a trial has randomized groups, the masking technique for groups, if the study includes a placebo group,

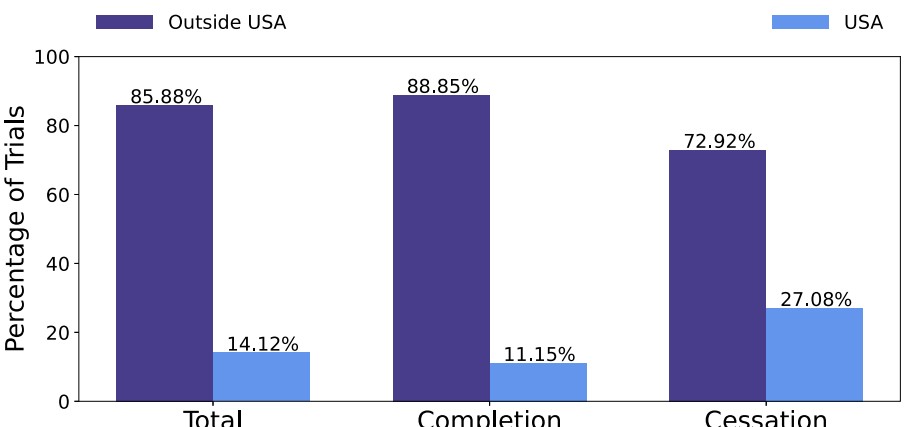

**Fig 6. Distribution of trials that are outside or inside USA with respect to all trials, successful trials and unsuccessful trials.**

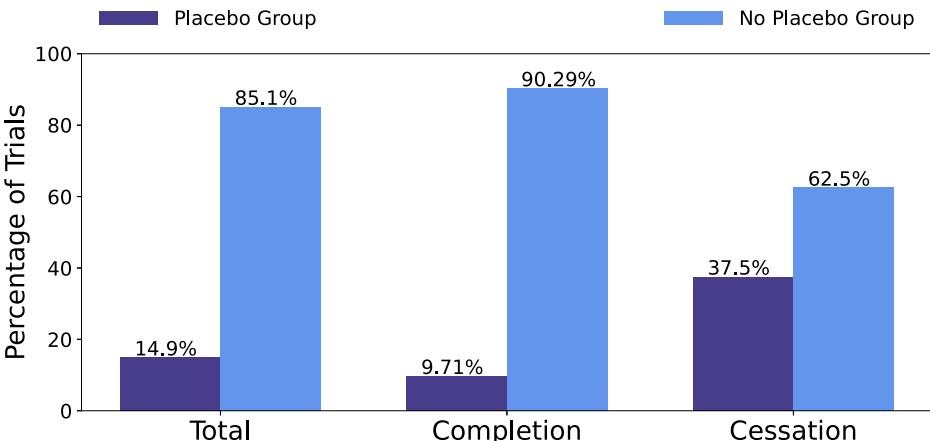

**Fig 7. Distribution of trials that have a placebo group respect to all trials, completion trials, and cessation trials, respectively.**

the number of groups, the number of countries with clinical sites and the total number of sites. These features indicate key aspects about the protocol used in the clinical trial. Increasingly complicated protocols, such as the masking technique and randomization of groups can add increasing complicated procedures that may hinder the success of a trial. However, study design aspects such as double blinding and randomised allocation of participants are required to estimate intervention effects [12].

While the administration of additional sites increases the financial burdens on the trial, increasing number of sites indicates that the trial has more access to enrolling participants. This is often necessary when the number of groups to enroll is increased. The usage of a placebo group is often required for medical interventional drugs to compare their intervention against a placebo control. However placebo groups indicate that the trial has to find additional participants. Trial can't meet their enrollment requirements will be ceased (terminated or withdrawn).

The distribution of placebo groups with regards to all trials, successful trials and unsuccessful trials is shown in Fig 7. Overall, a minority of clinical trials, 14.9%, in the data set use a placebo group. Of all cessation trials, 37.5% use a placebo group, which is five times higher than the percentage in the completion trials.

*Eligibility features.* Eligibility features record aspects about a trials eligibility requirements. Eligibility requirements are crucial to a clinical trial, because insufficient enrollment has been cited as a major contribution towards terminated clinical trials [8, 9]. Eligibility features include if the trial accepts healthy volunteers, has an age restriction, gender restriction and features created from the eligibility requirements field. The eligibility requirements field lists the inclusion and exclusion requirements for participants in the trial. A trial with too strict eligibility requirements has a high risk for insufficient enrollment. The eligibility field can be broken into separate lines that describe each eligibility requirement. Accordingly, features are created to count the number of inclusion and exclusion requirements. For trials do not have a distinction between inclusion and exclusion, the entire eligibility field is considered. Additionally, the number of words per inclusion, exclusion, and entire eligibility field are recorded, as well as the average words per inclusion/exclusion/eligibility line. Finally the number of numerical digits listed for inclusion/exclusion/entire eligibility requirements is also extracted. The number of numerical digits indicates strict requirements, such as the weight of a participant, which may become a more strenuous requirement to meet.

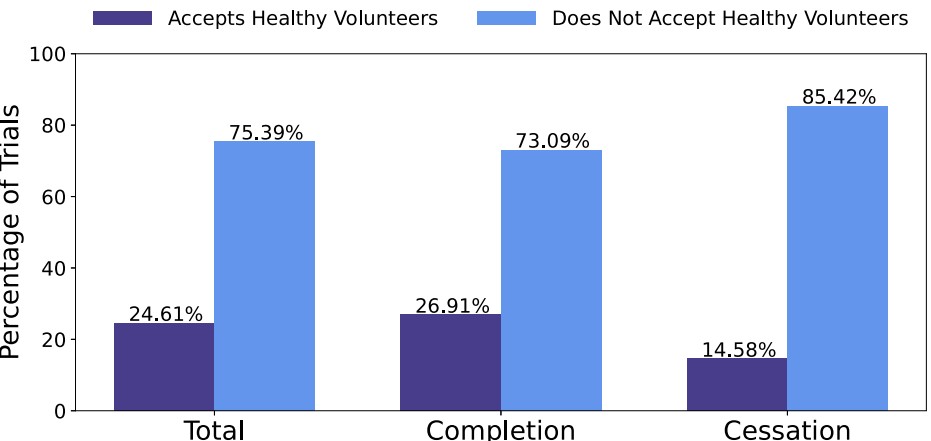

**Fig 8. Distribution of trials accepting healthy volunteers with respect to all trials, completion trials, and cessation trials, respectively.**

The distributions of clinical trials accepting healthy volunteers with respect to all trials, completion trials, and cessation trials are reported in Fig 8. COVID-19 Clinical trials may not accept healthy volunteers for multiple reasons. These might be observational trials for participants with COVID-19 or interventional trials that aim to study their intervention on participants currently diagnosed with COVID-19. Majority, 75.39%, of trials in the dataset do not accept healthy volunteers. For cessation trials, 85.42% do not accept healthy volunteers.

In S1 Table, we summarize all 40 statistics features and their brief descriptions.

**Drug features.**   As shown in Fig 5, while interventional trials make up 48.83% of all COVID-19 clinical trials in our dataset, 93.06% of cessation trials are interventional, which introduce a medical, device or even behavioral intervention to participants. Because drug development is an expensive and time-consuming endeavor, research efforts are motivated to re-purposing medical interventions to treat COVID-19. Many clinical trials have testing re-purposed drugs safety and efficacy for COVID-19 treatment. Some of these re-purposed drugs include Antivirals, Antimarial, Adjunctive pharmacological therapies and Corticosteroids [21]. While the hope that existing medical treatments are effective in treatment of COVID-19, not all of these treatments will provide positive results. Hydroxychloroquine, an antimalarial, was discontinued from one clinical trial due to no evidence of efficacy [21]. In the case where an intervention is noted as not safe or effective, the clinical trial will be terminated. Similarly, if a trial is planned to use a specific intervention that was proven as non-effective from separate sources or data, the planned trials will most likely terminate or withdraw as well.

Drug features are introduced to describe different interventions used to treat COVID-19. At a given time, there may be multiple concurrent trials researching similar drug interventions or drugs within the same drug class. The data that is reported about the safety and efficacy of these interventions may directly affect the outcome of concurrent trials. Trials may be ceased due to information from a different study or due directly to safety/efficacy concerns from their own study.

Two separate types of drug features are created in our study; (1) top drug features, and (2) top drug class features. These are derived from the Intervention Medical Subject Heading (MeSH) field in the clinical trial report. In total, we create 55 drug features, including 20 top drug features and 35 top drug class features.

*Top drug features*. The top 20 drug features are the 20 interventions with the most recorded clinical trials. The top 20 drug features in our dataset are listed in Table 3, which covers 218/

**Table 3. Top drug features and the number of clinical trials with the drug intervention.**

| Drug Feature | # Trials | Drug Feature | # Trials |
|---|---|---|---|
| Hydroxychloroquine | 73 | Interferons | 6 |
| Azithromycin | 30 | Prednisolone hemisuccinate | 5 |
| Ivermectin | 15 | Doxycycline | 5 |
| Ritonavir | 10 | Methylprednisolone Acetate | 5 |
| Lopinavir | 8 | Prednisolone | 5 |
| Dexamethasone | 7 | Prednisolone acetate | 5 |
| Chloroquine | 7 | IL-1Ra Protein | 5 |
| Methylprednisolone | 6 | Methylprednisolone Hemisuccinate | 5 |
| Antibodies | 6 | Prednisolone phosphate | 5 |
| Ascorbic Acid | 6 | Chloroquine diphosphate | 4 |

IL-1Ra represent Interleukin 1 Receptor Antagonist.

772 = 28.2% of all trials. To represent top drug interventions, we use one-hot encoding (*i.e.* a binary feature) to represent each drug feature. If a trial uses a specific drug intervention, the corresponding drug feature value is set as 1, or 0 otherwise.

In Fig 9, we report the distribution of Hydroxychloroquine with respect to different trials. Hydroxychloroquine is the most frequent drug intervention in the dataset, 9.46% of all COVID-19 clinical trials list Hydroxychloroquine as an intervention. For cessation trials, 31.94% of them use Hydroxychloroquine. The role of Hydroxychloroquine with COVID-19 has been an interesting one. Initial in vitro studies suggested that Hydroxychloroquine sulfate is a viable option for SARS-CoV-2 infections. This study was followed by a small observational trial that reported large clinical benefits. However, the study lacked key study design components, such as large sample size, randomisation and inclusion criteria. Initial media and political attention towards Hydroxychloroquine drove a large push for clinical trials to investigate the safety and efficacy [20]. As of June 2020, 88% of participants enrolled in ongoing clinical trials were those researching Hydroxychloroquine or Chloroquine [12]. Chloroquine is also an antimalarial intervention, both drugs belong to the drug class Aminoquinolines. As more evidence from clinical trials was analyzed, reports emerged that Hydroxychloroquine is neither effective nor safe for participants. These reports resulted in a large decrease in clinical trials

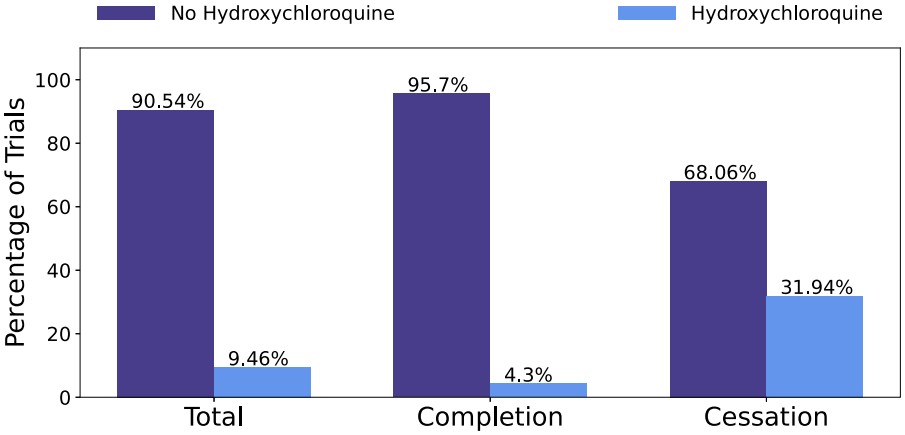

**Fig 9. Distribution of trials that used Hydroxychloroquine as an intervention.**

studying Hydroxychloroquine and some ongoing and pre-planned Hydroxychloroquine trials lost funding. Other trials are suspended or terminated due to the safety concerns [20]. This illustrates the importance of drug features. With limited resources and time, trials investigating interventions that are shown to be non-safe or ineffective from other ongoing research efforts will likely be ceased.

*Top drug class features*. Top drug features only capture most frequent drug intervention, and leave many unpopular drugs (the ones involving less than 5 trials) unrepresented. Therefore, we use drug class features to summarize all drug interventions. In total, there are 141 unique intervention MeSH terms, which can be grouped into respective drug classes, which are based on chemical substances or pharmacological action. Interventions belonging to the same drug class will effectively have a similar mechanism of action and often may have similar side effects or potential treatment power. Drug classes effectively summarize the similarity between multiple interventions and decrease the large feature space across all unique interventions.

For each intervention MeSH term, its drug class is found from NIH RxClass API [22], and the class names are taken from Anatomical Therapeutic Chemical (ATC) classification system. Drugs are classified based on Level 5, according to chemical substance. In the case where a drug intervention is not found from ATC level 5 classification, MeSHPA drug class is used instead. MeSHPA classifies drugs/chemicals based on MeSH terms that describe its pharmacological action (PA).

After obtaining drug classes for each drug intervention, the ones with at least 2 clinical trials in the drug class are used as top drug class features. From 141 drug interventions, there are 85 drug classes. Of these, 35 drug classes have interventions from 2 or more clinical trials. The top 10 drug classes, according to the number of clinical trials with interventions belonging to the class, are reported in Table 4.

After the 35 drug classes are found, for each clinical trial report, its top drug class features are created to represent the number of drug interventions the clinical trial has in each drug class, respectively.

**Table 4. Top 10 drug classes, the number of clinical trials with interventions in each class and the interventions belonging to the class.**

| Drug Class | # Trials | Drug Intervention(s) | | | |
|---|---|---|---|---|---|
| Aminoquinolines | 85 | Chloroquine | Primaquine | Hydroxychloroquine | Chloroquine diphosphate |
| Macrolides | 32 | Azithromycin | Clarithromycin | | |
| Corticosteroids acting locally | 18 | Prednisolone | Budesonide | Prednisone | Prednisolone phosphate |
| | | Prednisolone acetat | | | |
| Antivirals (HCV Treatment) | 16 | Ritonavir | Ribavirin | Sofosbuvir | Ledipasvir |
| | | Ledipasvir, Sofosbuvir drug combination | | | |
| Interferons | 16 | Interferons | Interferon-alpha | Interferon alpha-2 | Interferon Type I |
| | | Interferon-beta | Interferon-beta | Interferon beta-1a | |
| Other dermatologicals | 15 | Ivermectin | | | |
| Heparin group | 12 | Heparin | Calcium heparin | Tinzaparin | Dalteparin |
| | | Enoxaparin | | | |
| ARBs, plain | 10 | Losartan | Telmisartan | Candesartan | Olmesartan |
| | | Irbesartan | Eprosartan | | |
| Corticosteroids (local oral treatment) | 10 | Dexamethasone | Dexamethasone acetate | | |
| Antivirals (HIV Treatment) | 8 | Lopinavir | | | |

ARBs represent Angiotensin II receptor blockers.

**Keyword features.** Keyword features aim to capture important terms describing the clinical trial's summary. Within a clinical trial report, the keyword field has words or phrases describing the protocol. These are created using US National Library of Medicine (NLM) Medical Subject Heading (MeSH) terms. MeSH keywords are used for users to search for trials in the ClinicalTrials.gov database [19]. In some cases, the keyword field is missing. Thus keyword features are taken from the unique MeSH terms from the keywords field, condition MeSH field and condition field for each trial. This creates a string of keywords, separated with a comma, to describe each clinical trial. Keyword features are created from the Term-Frequency-Inverse Document Frequency (TF-IDF) of the keyword string. The string is tokenized into separated terms and the TF-IDF of each term is calculated. Token terms are taken by separating words by punctuation and spaces. Therefore, a multi-keyword term, like "Coronavirus Infections", would create multiple token terms, like "Coronavirus" and "Infections". After the term strings are tokenized, the TF-IDF score from each term is calculated. The top 500 terms from TF-IDF are used for keyword features, using their corresponding TF-IDF score.

TF-IDF weights the term-frequency of a term by its inverse document-frequency. Common terms appearing many times in a document will have a large term frequency $tf(t, T)$ but the weight value designated to them will be diminished by $idf(f)$, if the term frequently appears in many documents. Common terms (such as "the") often carry less meaningful information about the documents under study. In our research, each clinical trial report represents a document.

The TF-IDF of a keyword term, $f$, in a clinical trial report, $T$, is defined by $tf\text{-}idf(f, T)$, as shown in Eq (1), where the term-frequency, $tf(f, T)$, is the number of times term $f$ appeared in the clinical trial report $T$. This is weighted by the Inverse Document Frequency, $idf(f)$, as defined in Eq (2), where $df(f)$ represents the document frequency of term $f$, or the number of clinical trials that have the term; and $n$ represents the total number of clinical trials in the dataset ($n = 772$ in our study). The resulting TF-IDF values are then normalized by the Euclidean Norm.

$$tf - idf(f, T) = tf(f, T) \times idf(f) \tag{1}$$

$$idf(f) = \log \frac{1 + n}{1 + df(f)} + 1 \tag{2}$$

Table 5 reports the top 30 keyword terms, with respect to their average TF-IDF scores for all clinical trials. All clinical trials in the dataset represent those that are researching COVID-19, thus COVID-19 related terms are listed on the top; such as "Covid", "Coronavirus", "SARS" and "CoV". Since the MeSH keyword "SARS-CoV-2" is tokenized into two terms, "SARS" and "CoV", these are two separated keyword features. Other high ranked keyword terms indicate separated areas of research for individual clinical trials, such as "Depression"

**Table 5. The top 30 keywords with respect to average TF-IDF score.**

| Keywords | | | | | | | | | |
|---|---|---|---|---|---|---|---|---|---|
| Covid | Infection | Coronavirus | SARS | Pnemonia | CoV | Respiratory | Disease | Syndrome | Acute |
| Severe | Disorder | Virus | Hydroxychloroquine | Anxiety | Distress | Stress | Viral | Failure | Critical |
| Corona | Depression | Emergency | Psychological | Illness | Health | Lung | Ventilation | Care | ARDS |

ARDS represents Acute Respiratory Distress Syndrome.

**Table 6. The top 30 keywords with respect to average TF-IDF score for cessation trials, completion trials, and the keywords with the largest TF-IDF score difference between cessation *vs.* completion trials, respectively.**

| Trial Type | Keywords (ranked according to TF-IDF scores) | | | | | |
|---|---|---|---|---|---|---|
| Cessation | Covid | Coronavirus | Infection | SARS | CoV | Pneumonia |
| | Respiratory | Disease | Syndrome | Acute | Hydroxychloroquine | Severe |
| | Failure | Distress | Virus | Viral | Novel | Care |
| | Corona | Insufficiency | Patient | Prophylaxis | Lung | Healthcare |
| | Worker | Azithromycin | Tract | Cytokine | Pulmonary | Hypoxemia |
| Completion | Covid | Infection | Coronavirus | SARS | Respiratory | Pneumonia |
| | CoV | Syndrome | Disease | Acute | Severe | Disorder |
| | Anxiety | Stress | Virus | Distress | Critical | Depression |
| | Psychological | Illness | Emergency | Health | Corona | Viral |
| | Ventilation | Hydroxychloroquine | Lung | Treatment | Failure | ARDS |
| Difference | Covid | Coronavirus | SARS | CoV | Pneumonia | Infection |
| | Hydroxychloroquine | Respiratory | Disease | Anxiety | Failure | Critical |
| | Novel | Depression | Emergency | Psychological | Viral | Prophylaxis |
| | Azithromycin | Disorder | Distress | Syndrome | Worker | Diabetes |
| | Treatment | Healthcare | Tract | Mellitus | Care | Cytokine |

ARDS represents Acute Respiratory Distress Syndrome.

and "Anxiety". These trials are researching different health related fields with regards to COVID-19 participants.

To further understand keyword feature difference between completion *vs.* cessation trials, Table 6 lists the top 30 keyword features with respect to average TF-IDF score. The "Completion" and "Cessation" rows show high TF-IDF score terms in completion and cessation trials, respectively. The "Difference" row shows the top 30 keyword features that have the largest difference between average TF-IDF score of cessation trials compared to completion trials.

Table 6 shows that terms related to COVID-19 are listed on the top for both completion and cessation trials. Meanwhile, other terms may indicate the chance of success of trials with specific to research focus. For example, "Cytokine" is listed high for cessation trials, but not for completion trials. These trials are researching "Cytokine Storm" that is associated with COVID-19 infections. Cytokine storm refers to a hyperactive immune response of cytokines that are injurious to host cells. Cytokine storms have been associated with adverse outcomes in patients with severe COVID-19. Clinical trials that introduce monoclonal antibodies interventions often list Cytokine storm as their rationale [23]. These trials may be at a higher risk for unsuccessful outcomes due to their requirements for participants with high levels of cytokines or due to safety and efficacy concerns with the similar interventions under study for cytokine storm.

Terms such as "Depression" and "Anxiety" are ranked on the top list for completion trials. These trials may represent observational trials researching mental conditions of participants during the COVID-19 pandemic. As shown in Fig 5, observational trials have a lower rate of cessation outcome.

**Embedding features.** Each clinical trial report has a "Detailed Description" field providing textual descriptions of the clinical trial, such as the the objective of the clinical trials, the expected participates, the process and procedures etc. Since our keyword features already include top 500 terms, it will introduce significant redundancy to use Bag-of-words to represent such descriptions for prediction. Therefore, we propose to use embedding learning to

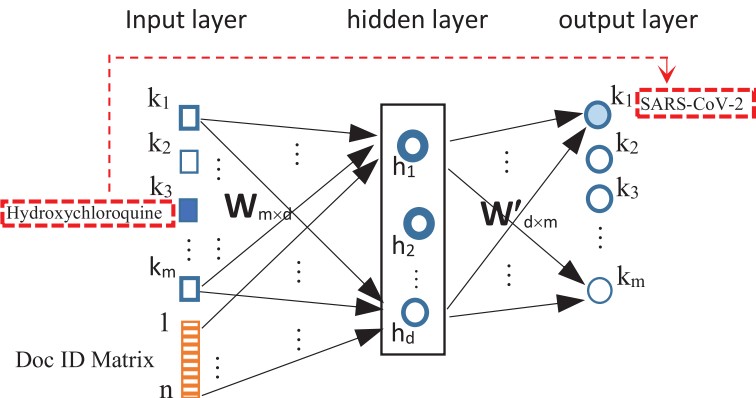

sentence: "hydroxychloroquine in intensive care patients infected with ``SARS-CoV-2"

**Fig 10. Doc2vec [25] neural network architecture for learning word and document embedding vector.** The input dimensions are *m* (corresponding to the vocabulary of documents) and *n* (corresponding the the number of documents). The hidden layer has *d* nodes, representing the embedding feature size. The output dimensions are *m* (corresponding to the vocabulary of documents). The principle is to learn to predict context. For example, if a report has "hydroxychloroquine" followed by "SARS-CoV-2" in the same sentence, the input corresponding to "Hydroxychloroquine" is set as one, and the Doc ID is also set as one. The network will learn weights to produce the largest output values corresponding to the node "SARS-CoV-2".

represent each document as a feature vector [24]. Comparing to using keywords to represent each clinical trial report as a bag-of-words, embedding features have the advantage that they are dense vectors (whereas bag-of-word features are sparse), and users can freely change the embedding feature size (we use 100 dimensions in the study).

To generate a feature vector to represent a clinical trial report, we use Doc2Vec [25], an expansion of the Word2Vec [26] using neural network language model to generate vector representations for words. Fig 10 shows the architecture of the network, which takes text as input to learn a feature vector to represent each word and each report. The input layer of the network corresponds to the vocabulary of all documents and the document IDs. The hidden layer has *d* neurons, whose weight values will be used as the embedding vector (*i.e.d* = 100 in our study). The output layer maps to the vocabulary of all documents. The embedding learning is to train the network to predict surround words. For example, for a report sentence, such as "Hydroxychloroquine in intensive care patients infected with "SARS-CoV-2", because "Hydroxychloroquine" and "SARS-CoV-2" both occur in the sentence, the input corresponding to "Hydroxychloroquine" is set as one, and train network weights such that the output node corresponding to "SARS-CoV-2" has the largest output.

In order to learn a vector representation for each clinical trial, each report is also added as an extra input to the network. After training the model using all reports, the *d* dimensional weight values connecting each report to the hidden nodes are used as the embedding features. In our study, the Doc2Vec model creates a vector of length 100 to represent each detailed description.

**Feature selection.** The above feature engineering efforts create 693 features in total to represent each clinical trial. In order to understand which features are most informative towards the trial completion or cessation, we use ReliefF feature selection method [27] to study feature impact.

ReliefF is a similarity based feature selection method. It evaluates quality of each feature, *f*, on its capability to differentiate instances near to each other. Given a randomly selected instance, $\mathbf{x}_i$, ReliefF first finds its nearest hit set $\mathcal{H}_i$, which includes *k* nearest neighbors with

the same class label as $\mathbf{x}_i$, and nearest misses set $\mathcal{M}_i$ including $k$ nearest neighbors with different class label from $\mathbf{x}_i$.

For each feature $f$, its weight value $w[f]$ is initially set to 0, and updated based on Eq (3). The motivation is that if the feature $f$ has different values from two instances in the same class, the weight value is decreased, since the feature does not convey similarity between same class samples. However if the feature has different values from two instances from different classes, the feature differentiates between opposite classes, and the features quality score increases.

According to Eq (3), for each instance $\mathbf{x}_j \in \mathcal{H}_i$ in the nearest hit set of instance $\mathbf{x}_i$, feature $f$'s weight value is decreased based on the difference score between $\mathbf{x}_i$ and $\mathbf{x}_j$, diff($f$, $\mathbf{x}_i$, $\mathbf{x}_j$), divided by $m \times k$ where $m$ represents the number of features in the dataset. On the other hand, the weight value $w[f]$ is increased by the difference between $\mathbf{x}_i$ and each miss neighbor $\mathbf{x}_j \in \mathcal{M}_i$. The contribution of miss class is weighted with prior probability of miss class $C$, $P(C)$, divided by $1 - P(\mathbf{x}_i)$ [27].

$$
\begin{aligned}
w[f] \quad &= w[f] - \sum_{j=1, x_j \in \mathcal{H}_i}^{k} \frac{\mathrm{diff}(f, x_i, x_j)}{(m \times k)} \\
&+ \sum_{C \neq class(x_i)} \left( \frac{\left[ \frac{P(C)}{1-P(class(x_i))} \sum_{j=1, x_j \in \mathcal{M}_i}^{k} \mathrm{diff}(f, x_i, x_j) \right]}{(m \times k)} \right)
\end{aligned}
\tag{3}
$$

The function diff($f$, $\mathbf{x}_i$, $\mathbf{x}_j$) calculates the difference between two instances, $\mathbf{x}_i$ and $\mathbf{x}_j$, with respect to feature $f$. For nominal features, this is calculated using Eq (4), and for continuous features, this is calculated using Eq (5) [27], where $x_{i,f}$ denotes $\mathbf{x}_i$'s value in feature $f$.

$$
\mathrm{diff}(f, \mathbf{x}_i, \mathbf{x}_j) = \begin{cases} 0, & \text{if } x_{i,f} = x_{j,f} \\ 1, & \text{otherwise} \end{cases}
\tag{4}
$$

$$
\mathrm{diff}(f, \mathbf{x}_i, \mathbf{x}_j) = \frac{|x_{i,f} - x_{j,f}|}{max(f) - min(f)}
\tag{5}
$$

After ReliefF is applied, the end result is a ranking of features based on their weight scores. For the top 50, 100, 200, 300, 400, 500, 600 and all features, the percentages of statistics, Keyword, top drug, drug class, and embedding features are reported in Fig 11.

The results in Fig 11 show that, in terms of importance, the four types of features are ranked in a descending order, from keyword features, drug features (including top drug class features and top drug features), statistics features, to embedding features. When only a small number of features (*e.g.* 50) are selected, the selected features mainly include keywords (76%), drug features (22%), and statistics features (2%). Overall, embedding features are least informative. This is different from our previous study [10], where statistics features are found to be more informative than keyword features and embedding features. Nevertheless, our experiments in the next section will show that, despite of the individual importance, all four types of features contribute to the accurate prediction of COVID-19 trials. The models using all four types of features often have the best performance.

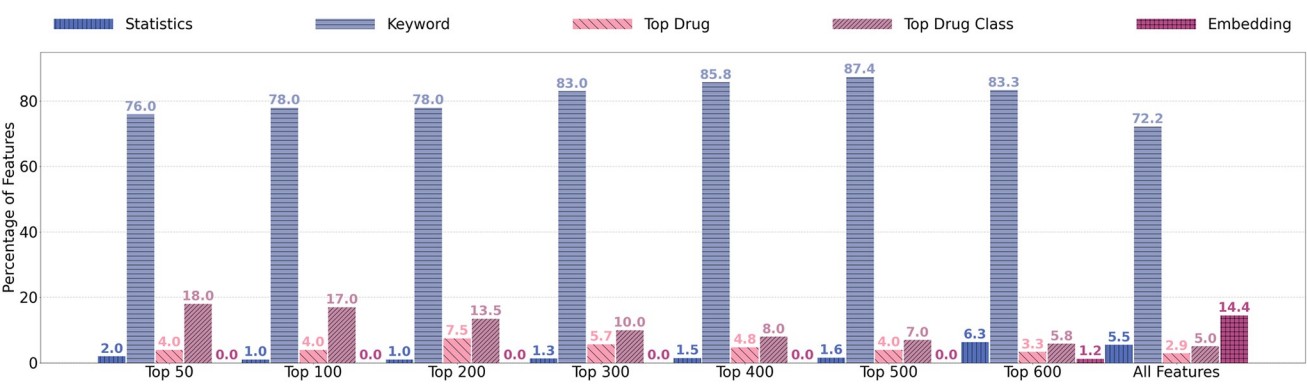

**Fig 11. Percentage of types of features within the top ranked 50–600 and all features.**

## COVID-19 clinical trial prediction

**Prediction framework.** For the selected 772 COVID-19 clinical trials, we first create statistics features, keyword features, and drug features to represent each trial. The dataset is separated into five subsets of training and test samples for 5-fold cross validation. Doc2Vec is trained on each training dataset and the embedding vector is inferred for each training and test dataset. In addition, feature selection or ranking is also carried out by using training data only. The reason of using this step is to ensure that test data are completely held out during the learning process, so we can validate the genuine model performance without knowing any of the test data information.

*Ensemble learning.* Our research will study a variety of predictive models for trial prediction. Meanwhile, in order to exploit the best possible performance, we will employ ensemble learning to combine multiple models. Because there are only 144 cessation trials, out of all 772 trials, this results in class imbalance in the training set (*i.e.* completion trials dominate the whole dataset). Therefore, random under sampling is applied to completion trials to produce a sampled set relatively balanced to the number of cessation trials. In the original dataset, the imbalance ratio between completion trials to cessation trials is 4.36 to 1. By using random under sampling, we will vary the class distribution from 4.36:1 to 4:1, 3:1, 2:1, 1:1, and 0.9:1. This allows our study to understand best under sampling ratios in order to achieve the best model performance.

Because random under sampling may potentially remove important examples and introduce bias to the trained models, the sampling are repeated 10 times, with each sampled data set producing one predictive model. The 10 trained models are then combined to form an ensemble to predict each test trial.

**Predictive models.** For comparison purposes, we use four models, Neural Network, Random Forest, XGBoost and Logistic Regression, in our study for COVID-19 trial prediction.

*Neural network.* Our neural network implementation is a three layer feed forward neural network, consisting of an input layer, one hidden layer with 50 nodes and an output layer with a single node to produce the classification label (completion trial or cessation trial). Neural network hyper parameters are optimized using an exhaustive nested 5-fold grid search. The grid search determines a batch size of 50, 50 training epochs, 50 nodes and the sigmoid activation function as optimal parameters. The *Adam* optimization function [28] is used during the back propagation phase.

*Random forest.* Random forests are an ensemble of decision tree classifiers. Random forest hyper parameters are optimized using an exhaustive nested 5-fold grid search. The final

random forest implementation does not use bootstrap samples, but uses the whole dataset to build each tree (using random feature subsets). The entropy criterion is used to measure the quality of split. The number of features to consider for the best split is set to $\sqrt{m}$, where $m$ is the number of features in the dataset. A minimum of five samples are set to split each node. Trees are grown until there are at minimum two samples in each leaf node. The random forests consist of 1,500 decision trees.

*XGBoost.* Extreme Gradient Boosting, XGBoost, is an ensemble decision tree algorithm with gradient boosting framework. In order to optimize XGBoost, we use an exhaustive nested 5-fold grid search to find final hyper parameters. The final parameters use 600 trees. The minimum loss reduction is set to two; maximum depth is five; minimum weight in a child node is one; feature sub-sampling ratio is 0.6; sampling sub-sampling ratio is one (no sub-sampling on training samples).

*Logistic regression.* Logistic Regression is a nonlinear classification model, using a logistic function defined in Eq (6), to model probability of a trial $\mathbf{x}_i$ belonging to a positive ($y = 1$ meaning cessation trial) or negative ($y = -1$ meaning completion) class, and $\mathbf{w}$ is the weight vector [29].

$$\mathcal{P}_{\mathbf{w}}(y = \pm 1 | \mathbf{x}) = \frac{1}{1 + e^{-y \mathbf{w}^T \mathbf{x}_i}} \tag{6}$$

In order to learn weight values $\mathbf{w}$, a binary class $l_2$ penalized normalization is used to minimize the cost function in Eq (7), where $C > 0$ is a penalty parameter [29].

$$\operatorname*{argmin}_{\mathbf{w}} \mathcal{P}(\mathbf{w}) = C \sum_{i=1}^{n} \log\left(1 + e^{-y \mathbf{w}^T \mathbf{x}_i}\right) + \frac{1}{2} \mathbf{w}^T \mathbf{w} \tag{7}$$

**Performance metrics.** To asses the performance of these models, we consider Accuracy, Balanced Accuracy, F1-score and AUC. The dataset is imbalanced, the ratio between completion trials to cessation trials is 4.36 to 1, thus balanced accuracy can be a better indicator of model performance. Balanced accuracy is the average of accuracy of each individual class.

**Statistical tests.** A Friedman test is used to determine if is there is a statistical difference between the four models trained on different combinations of features, which are treated as separate data sets [30]. Each classifier produces an average performance metric for each data set. These performance metrics are then ranked in a descending order to evaluate the classifiers, so the classifier with the highest scores on a dataset is ranked as 1. In the case of a tie, the average rank is assigned. Then the average ranking of four classifiers are determined over the seven combinations of features. The average rank of a classifier $j$ is denoted as $R_j$ as defined in Eq (8), where $r_i^j$ is the rank for classifier $j$ on dataset $i$. In our analysis, we have four classifiers and seven data sets. The data sets are those created using top 100, 200, 300, 400, 500, 600 features, and all features, respectively.

$$R_j = \frac{1}{n} \sum_{i=1}^{n} r_i^j \tag{8}$$

The null hypothesis states that there is no difference between algorithms, thus their average ranks is not statistically different. The Friedman statistic is defined by $\chi_F^2$, in Eq (9), where $k$

denotes the number of classifiers.

$$\chi^2_F = \frac{12n}{k(k+1)} \left[ \sum_{j=1}^{k} R_j^2 - \frac{k(k+1)^2}{4} \right] \qquad (9)$$

After rejecting the null-hypothesis that the classifiers are equivalent, a Nemenyi post-hoc test can be performed for pairwise comparisons of performance. Classifiers are significantly different if their average ranks differ by the critical difference, *CD* as defined in Eq (10), where $q_\alpha$ is the Studentized range statistic divided by $\sqrt{2}$ [30]. In our study, we have $k = 4$ classifiers and $\alpha = 0.05$, $q_{0.05} = 2.569$, and for a lower level of confidence, $\alpha = 0.10$, $q_{0.10} = 2.291$. Since there are $n = 7$ data sets, with $\alpha = 0.05$, $CD = 1.773$, and with $\alpha = 0.10$, $CD = 1.581$.

$$CD = q_\alpha \sqrt{\frac{k(k+1)}{6n}} \qquad (10)$$

The Nemenyi post-hoc test results, showing in a critical difference diagram, are reported in Fig 15. The top line indicates the average ranks $R_j$ of each classifier. The classifiers with the highest performance and lowest ranks are displayed on the left, and classifiers with the lowest performance and highest ranks are displayed on the right. Any two classifiers that are not significantly different have average ranks that do not differ by the *CD* and are grouped together with a bar.

## Results

### Feature selection for COVID-19 trial prediction results

In order to determine whether ReliefF feature selection helps COVID-19 clinical trial prediction, four models are tested on different feature subsets. The top ranked 100, 200, 300, 400, 500, 600, and all features are tested separately. The average AUC values from 5-fold cross validation on the different feature subset are shown in Fig 12.

Generally, as the number of features increases, the AUC increases. Using all features has similar, if not slightly decreased, results from using the top 600 ranked features. Embedding features are ranked as the 100 lowest using ReliefF feature selection. The results from Fig 12 suggest that embedding features are the least helpful to increase prediction performance. This is in contrast to our previous findings that the combination of embedding features, keyword features, and statistics features provide the highest predictive power [10].

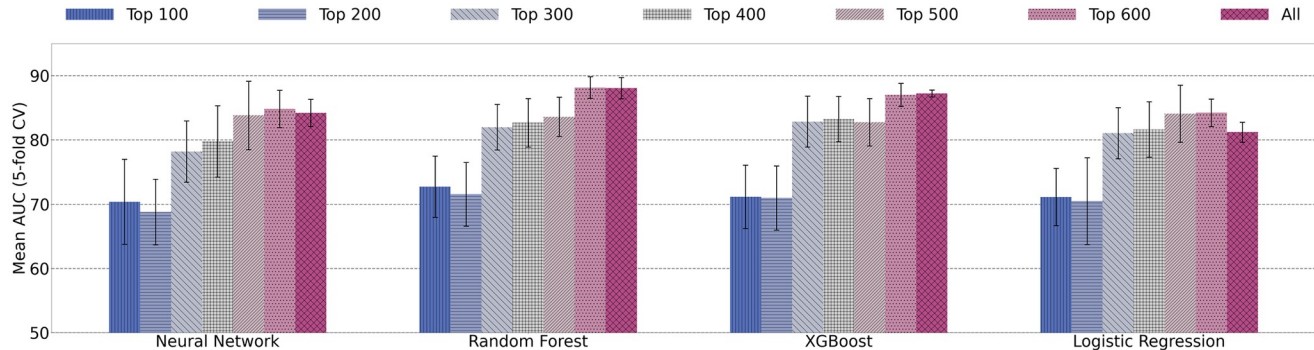

**Fig 12. Mean AUC values and standard deviation of single models trained using increasing number of features.**

## Sampling ratio for COVID-19 trial prediction results

Our previous research demonstrated that ensemble models are able to handle the class imbalance in clinical trials. Thus for the current study, we use random under sampling to the negative class (completion trials) to achieve a more even class proportion to cessation trials. Since this introduces a bias, it is repeated 10 times to create an ensemble model, and the average of all models is used for final test predictions. In order to find best sampling ratios, Fig 13 demonstrates average accuracy, balanced accuracy, F1-score, and AUC for the four models using no random under sampling and sampling rates of 4, 3, 2, 1 and 0.9 to 1. The sampling rate of 2:1 indicates that the ratio between number of negative class samples (completion trials) to positive class samples (cessation trials) is 2:1. A sampling rate of 1:1 indicates an even ratio and 0.9:1 indicates that there are more cessation trials than completion trials.

As the sampling rate decreases, the overall accuracy rate of the models decrease, while balanced accuracy and F1-score increase. As the class ratios become even, the models increasingly classify more samples as cessation trials. This increases the number of true positives (*i.e.*, cessation trials are correctly predicted as cessation trials), while concurrently increases the number of false positives (*i.e.*, completions trials are predicted as cessation trials). This ultimately will increase the balanced accuracy rate as more samples from both classes are classified correctly. However, the accuracy rate will decline as more completion trials are incorrectly classified. As F1 score and AUC do not differ greatly between 1:1 sampling and 0.9:1 sampling, we use 1:1

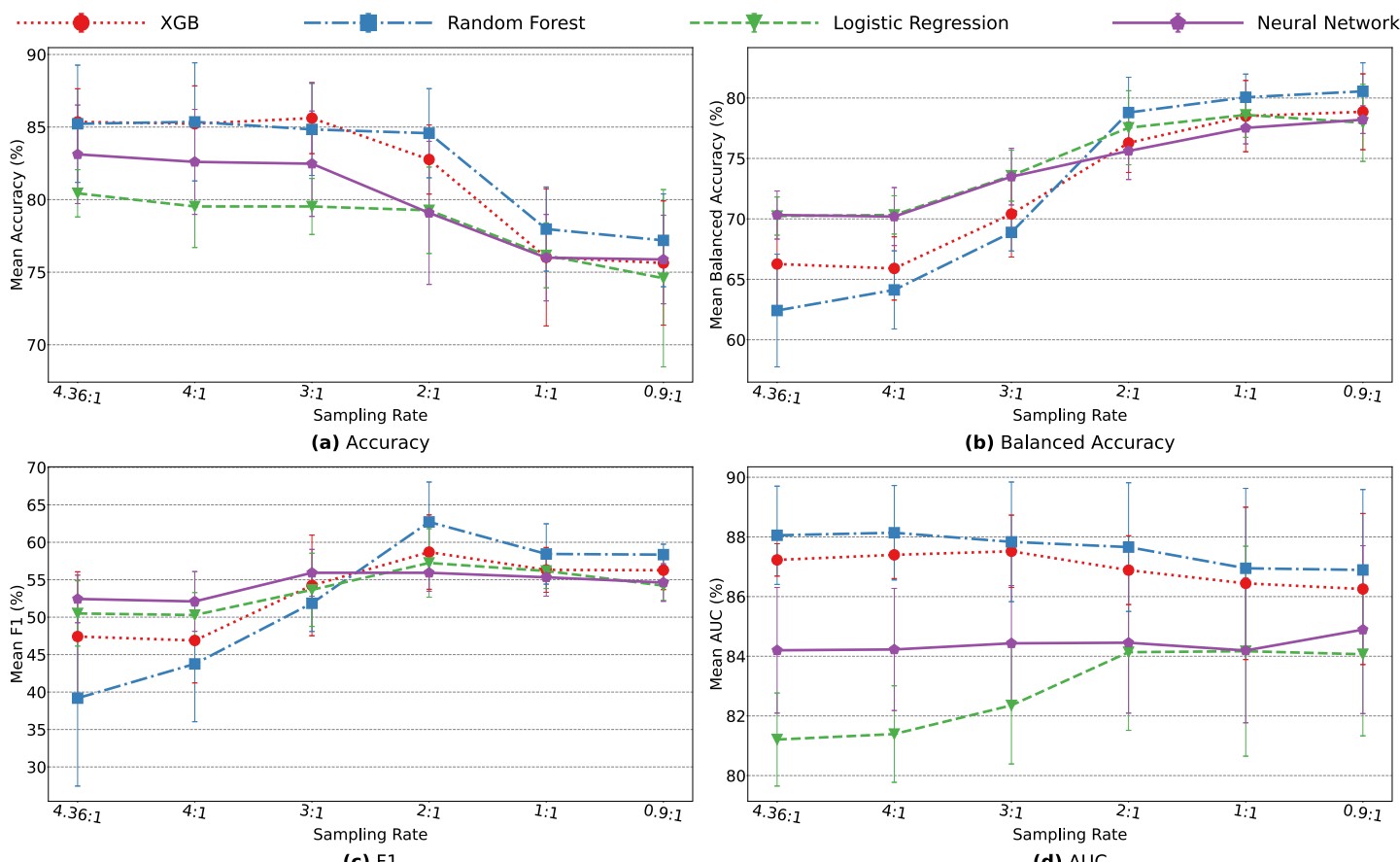

**Fig 13. Mean accuracy, balanced accuracy, F1-score and AUC values comparisons using different sampling ratios.** 4.36:1 indicates no random under sampling (original sample distributions between completion trials *vs.* cessation trials in the dataset).

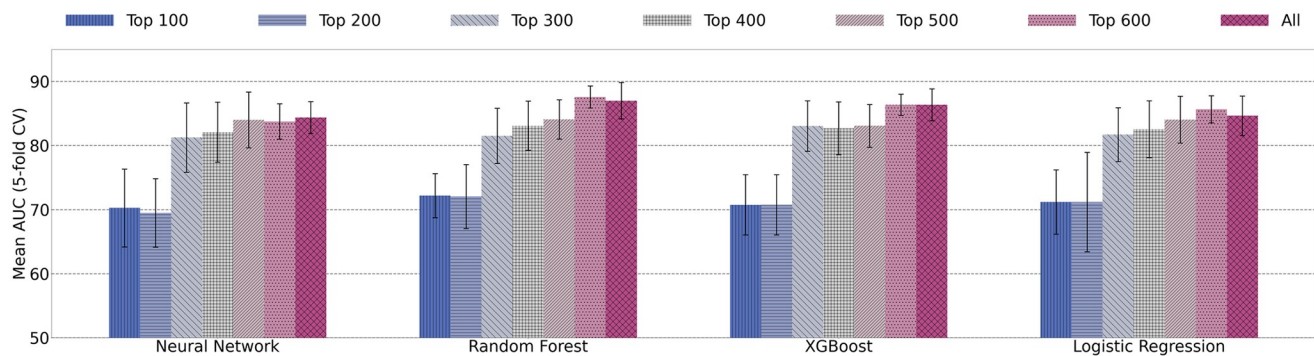

**Fig 14. Mean AUC values and standard deviation of ensemble models trained using increasing number of features.**

sampling rate for our final ensemble models. This still shows an increase of balanced accuracy without the largest decrease of overall accuracy.

## Ensemble learning for COVID-19 trial prediction results

For ensemble models, to determine if ReliefF feature selection aids in prediction power, we test ensemble models on subsets of features from their ReliefF ranking. Fig 14 demonstrates 5-fold cross validation AUC on ensemble models using the top 100, 200, 300, 400, 500, 600 ranked features, and all features. Similar to the single model results, the best performance is seen with top 600 features or using all features. The difference between using top 600 and all features is minor. For example, random forest with top 600 features has AUC score of 87.58% *vs.* AUC of 86.99% using all features. In addition, we also report the performance between single models *vs.* ensemble models using all features and using top ranked 600 features in Table 7.

Overall, the results in Fig 14 show that feature selection does not improve AUC scores for trial prediction. When a small number of features are selected (*e.g.* less than 100), the models appear to deteriorate significantly. With increasing number of features being used in the classification models, AUC performance increases. This confirms that all four types of features (statistics features, keyword features, drug features, and embedding features) play important roles and provide complementary information for prediction.

Table 7 list the average 5-fold cross validation performance of single models *vs.* ensemble models trained using all features. The results show a decrease of accuracy for ensemble models; while F1, Balanced Accuracy and AUC increase. Ensemble models utilize sampling to balance the sample distributions in the training set. This ultimately will increase the number of cessation clinical trials predicted correctly, and concurrently will increase the number of completed clinical trials predicted incorrectly, which decreases overall accuracy.

To compare the performance between ensemble classifiers tested on different combinations of features, a Friedman test shows a significant difference for Balanced accuracy, F1, and AUC

**Table 7. Average performance comparisons between single models *vs.* ensemble models using all features.**

|  | Single Models | | | | Ensemble Models | | | |
|---|---|---|---|---|---|---|---|---|
|  | Accuracy | Balanced | F1 | AUC | Accuracy | Balanced | F1 | AUC |
| Neural Network | 83.12% | 70.32% | 52.43% | 84.20% | 75.97% | 77.53% | 55.37% | 84.37% |
| Random Forest | 85.23% | 62.42% | 39.19% | 88.05% | 78.62% | 81.21% | 59.75% | 86.99% |
| XGBoost | 85.35% | 66.27% | 47.42% | 87.23% | 76.80% | 78.60% | 56.83% | 86.35% |
| Logistic Reg. | 80.43% | 70.23% | 50.49% | 81.21% | 76.29% | 78.43% | 56.15% | 84.65% |

(a)

(b)

(c)

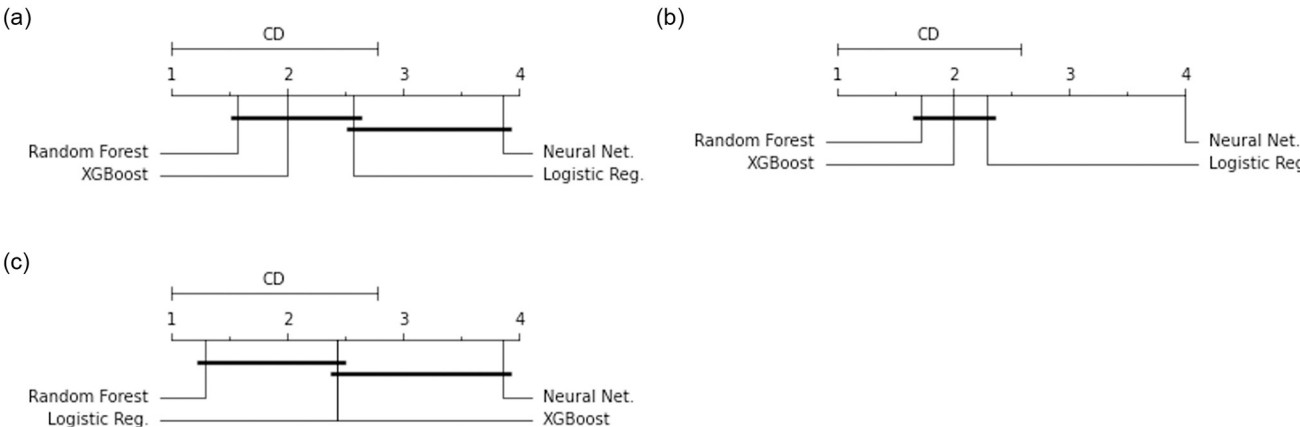

**Fig 15. Critical difference diagram for ensemble models by comparing the four classifiers on different combinations of features; (a) Balanced Accuracy scores with $\alpha = 0.05$, the critical difference is 1.77; (b) F1 with $\alpha = 0.1$, the critical difference is 1.58; (c) AUC with $\alpha = 0.05$, the critical difference is 1.77.** Groups of classifiers not significantly different are connected.

scores. There is no statistical difference between accuracy scores. Fig 15 displays a critical difference diagram demonstrating the Nemenyi post-hoc test results. For balanced accuracy, Friedman test demonstrates a significant difference between the four classifiers, $\chi^2_F = 12.429$, $p = 0.006$. The Nemenyi post-hoc test results using $\alpha = 0.05$ in Fig 15 (a) show that Random Forest and XGBoost are significantly better than Neural Network. For F1 scores, a Friedman test demonstrates a significant difference between the four classifiers, $\chi^2_F = 13.286$, $p = 0.004$. The Nemenyi post-hoc test results using $\alpha = 0.1$ in Fig 15 (b) show that Random Forest, XGBoost and Logistic Regression are significantly better than Neural Network. For AUC scores, a Friedman test demonstrates a significant difference between the four classifiers, $\chi^2_F = 13.971$, $p = 0.003$. The Nemenyi post-hoc test results using $\alpha = 0.05$ in Fig 15 (c) show that Random Forest is significantly better than Neural Network.

Overall, the critical difference diagrams in Fig 15 demonstrate that Random Forest has the best average ranking of model performance in Balanced Accuracy, F1, and AUC scores. While Neural network has the lowest ranking in all three performance metrics. These statistical results indicate two major findings; (1) Random Forest achieves the highest performance; and (2) Neural network has the lowest performance.

Our results can be summarized into the following major findings:

- Feature selection does not improve AUC scores for trial prediction.

- Ensemble models show an improvement in F1 and balanced accuracy scores, a minimal improvement in AUC scores, and a decrease in accuracy scores.

- Random forest has the best performance and neural network has the worst performance.

- Random Forest ensemble models achieve 81% in balanced accuracy and 87% in AUC, indicating our model can achieve satisfactory accuracy for completion *vs.* cessation COVID-19 clinical trial prediction.

## Discussion

Using feature engineering and predictive modeling, our research shows much higher AUC values and balanced accuracy in predicting COVID-19 clinical trials, comparing to our previous study [10] which focused on all clinical trials. Because COVID-19 clinical trials all aim at

researching COVID-19 (*i.e.* a particular disease), these trials share more common features comparing to all clinical trials. The segregation of clinical trials based on research area shows a large improvement in predictive power of modeling clinical trials. Previously, we demonstrated single models that achieved only 50% balanced accuracy and F1-score of 0–2%, where a whole class of samples were largely missclassified. Our current study shows single models with balanced accuracy as high as 70% and F1-score of 50.49%. This observation suggests that modeling clinical trials is best when segregating research areas or diseases.

The high ranking of drug features from ReliefF feature selection demonstrates their ability to separate cessation *vs.* completion trials. Since the majority of COVID-19 clinical trials are interventional, it's understandable that drug interventions will be important key components of a trial's success.

The low ranking of embedding features from ReliefF suggest that the embedding feature values are not as separable when comparing the two classes. This could be due to the large similarities between the description fields of all COVID-19 clinical trials. Descriptions for COVID-19 clinical trials may contain similar background information necessary for justification of the trial. This may include details on the emergence of COVID-19 or the consequences of the global pandemic. Finally the dataset is considerably smaller than our previous research. The Doc2Vec models are only trained on the training datasets. With a smaller datasets, Doc2-Vec has less samples to properly learn an embedding model.

## Conclusion

This paper proposes to study factors associated to the COVID-19 clinical trial completion *vs.* cessation, and further employs predictive models to predict whether a COVID-19 clinical trial may complete or cease in future. Four types of features (statistics features, keyword features, drug features, and embedding features) with 693 dimensions are created to represent each clinical trial. Feature selection and ranking show that keyword features (which are derived from the MeSH terms of the clinical trial reports) are most informative for COVID-19 trial prediction, followed by drug features, statistics features, and embedding features. Using random under sampling and ensemble learning, the results show that ensemble models achieve significant gain in balanced accuracy and F1-score. Overall, the best models achieve over 0.87 AUC values and over 0.81 balanced accuracy for completion *vs.* cessation trial prediction.

A limitation of this study is that it is based on a rather small number of clinical trial reports (772) which had reported completion or cessation (terminated, suspended, or withdrawn) status in January 2021. As COVID-19 clinical research advances rapidly, more trials will have updated status, and may contribute to the study. In addition, other machine learning models, such as semi-supervised models, can also be integrated to combine COVID-19 trials not reaching the completion or cessation states to improve the model performance.

## Supporting information

**S1 Table. Statistics features.** The 40 statistics features, their subcategory and definition. (PDF)

## Author Contributions

**Data curation:** Magdalyn E. Elkin, Xingquan Zhu.

**Formal analysis:** Xingquan Zhu.

**Funding acquisition:** Xingquan Zhu.

**Investigation:** Xingquan Zhu.

**Methodology:** Magdalyn E. Elkin, Xingquan Zhu.

**Project administration:** Xingquan Zhu.

**Resources:** Magdalyn E. Elkin.

**Software:** Magdalyn E. Elkin.

**Supervision:** Xingquan Zhu.

**Validation:** Magdalyn E. Elkin, Xingquan Zhu.

**Writing – original draft:** Magdalyn E. Elkin.

**Writing – review & editing:** Xingquan Zhu.

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
