## [Decision Letter · Decision Letter 0]

4 May 2021

PONE-D-21-11234

Understanding and Predicting COVID-19 Clinical Trial Completion vs. Cessation

PLOS ONE

Dear Dr. Zhu,

Thank you for submitting your manuscript to PLOS ONE. After careful consideration, we feel that it has merit but does not fully meet PLOS ONE’s publication criteria as it currently stands. Therefore, we invite you to submit a revised version of the manuscript that addresses the points raised during the review process.

Based on the comments received from the reviewers and my own observation, I suggest major revisions for the paper.

We look forward to receiving your revised manuscript.

Kind regards,

Thippa Reddy Gadekallu

Academic Editor

PLOS ONE

Journal Requirements:

Reviewers' comments:

Reviewer's Responses to Questions

**Comments to the Author**

1. Is the manuscript technically sound, and do the data support the conclusions?

Reviewer #1: Yes

Reviewer #2: Partly

2. Has the statistical analysis been performed appropriately and rigorously? 

Reviewer #1: Yes

Reviewer #2: No

3. Have the authors made all data underlying the findings in their manuscript fully available?

Reviewer #1: Yes

Reviewer #2: No

4. Is the manuscript presented in an intelligible fashion and written in standard English?

Reviewer #1: Yes

Reviewer #2: No

5. Review Comments to the Author

Reviewer #1: 1. In Introduction section, the drawbacks of each conventional technique should be described clearly.

2. Introduction needs to explain the main contributions of the work more clearly.

3. The authors should emphasize the difference between other methods to clarify the position of this work further.

4. The Wide ranges of applications need to be addressed in Introductions

5. The objective of the research should be clearly defined in the last paragraph of the introduction section.

6. The authors can cite the following references.

Analysis of dimensionality reduction techniques on big data.

Deep learning and medical image processing for coronavirus (COVID-19) pandemic: A survey.

An Incentive Based Approach for COVID-19 planning using Blockchain Technology.

Reviewer #2: Please re structure the paper format

1. What are the limitations of the existing literature?

2. List out the main contributions of the current work. Visit this profile and cite couple of related papers: https://scholar.google.com/citations?user=dz4QKZIAAAAJ&hl=en

3. There are many typos and grammatical errors in the paper. The authors have to carefully proofread the paper to address them.

4. Summarize the related works section with a table.

5. Present a detailed analysis of the results obtained.

6. Discuss about the limitations and future scope of the present work.

6. PLOS authors have the option to publish the peer review history of their article (what does this mean?). If published, this will include your full peer review and any attached files.

Reviewer #1: No

Reviewer #2: No

---

## [Author Response · Author response to Decision Letter 0]

28 May 2021

The major changes of this revision are summarized as follows:

• We added additional information to the Introduction section and ensured a clear statement of our study objective.

• A new subsection Related Work was added to the Introduction to discuss related research and compare the existing research to our current study. 

• Table 1 was added in subsection Related Work to present a graphical summary and comparison between existing methods and proposed research.

• We have added a subsection Contribution to the Introduction to summarize our current study's contribution. The main contributions of our study are as follows:

o COVID-19 clinical trial benchmark: Our research delivers a COVID-19 benchmark for clinical trial completion vs. cessation study. The benchmark, including features and supporting documents, are published online for public access. 

o Clinical trial features: Our research proposes the most extensive set of features for clinical trial reports, including features to model trial administration, study information and design, eligibility, keywords, and drugs etc. In addition, our research also uses embedding features to model unstructured free texts in clinical trial reports for prediction. 

o Predictive modeling of COVID-19 trials: Our research is the first effort to model COVID-19 clinical trial completion vs. cessation. By using ensemble learning and sampling, our model achieves over 0.87 AUC scores and over 0.81 balanced accuracy for prediction, indicating high efficacy of using computational methods for COVID-19 clinical trial prediction. 

• In the revision, we introduced Friedman statistical tests to determine the statistical difference between ensemble classifiers tested over the different combinations of features. To give background information on the statistical test, a subsection Statistical Tests was added to the COVID-19 Clinical Trial Prediction section to explain the Friedman test and Nemenyi post hoc test.

• Results from Friedman test and Nemenyi post hoc test are reported in COVID-19 Clinical Trial Prediction section. Figure 15 was added to show the critical difference diagrams for ensemble models' Balanced accuracy, F1, and AUC scores.

• To simplify the results, we only display average performance comparisons between single models vs. ensemble models using all features. 

• The last paragraph of the result section presents an itemized list to clearly summarize the major findings from our experimental results. 

• We have gone carefully through the paper and fixed spelling and grammatical errors.

In the revision summary, we provide detailed responses to all the comments from the reviewers. The text in italics is taken verbatim from the reviewers’ comments, and the roman text is our response.

---

## [Decision Letter · Decision Letter 1]

14 Jun 2021

Understanding and Predicting COVID-19 Clinical Trial Completion vs. Cessation

PONE-D-21-11234R1

Dear Dr. Zhu,

We’re pleased to inform you that your manuscript has been judged scientifically suitable for publication and will be formally accepted for publication once it meets all outstanding technical requirements.

Kind regards,

Thippa Reddy Gadekallu

Academic Editor

PLOS ONE

Additional Editor Comments (optional):

Reviewers' comments:

Reviewer's Responses to Questions

**Comments to the Author**

1. If the authors have adequately addressed your comments raised in a previous round of review and you feel that this manuscript is now acceptable for publication, you may indicate that here to bypass the “Comments to the Author” section, enter your conflict of interest statement in the “Confidential to Editor” section, and submit your "Accept" recommendation.

Reviewer #1: All comments have been addressed

Reviewer #2: (No Response)

2. Is the manuscript technically sound, and do the data support the conclusions?

Reviewer #1: Yes

Reviewer #2: (No Response)

3. Has the statistical analysis been performed appropriately and rigorously? 

Reviewer #1: Yes

Reviewer #2: (No Response)

4. Have the authors made all data underlying the findings in their manuscript fully available?

Reviewer #1: Yes

Reviewer #2: (No Response)

5. Is the manuscript presented in an intelligible fashion and written in standard English?

Reviewer #1: (No Response)

Reviewer #2: (No Response)

6. Review Comments to the Author

Reviewer #1: I have gone the revised paper, the authors have addressed all my comments, paper can be accepted in the current form

Reviewer #2: Author has submitted all the recommendations

I am happy to accept the paper for publication.

7. PLOS authors have the option to publish the peer review history of their article (what does this mean?). If published, this will include your full peer review and any attached files.

Reviewer #1: No

Reviewer #2: No

---

## [Editor Report · Acceptance letter]

29 Jun 2021

PONE-D-21-11234R1 

Understanding and Predicting COVID-19 Clinical Trial Completion *vs.* Cessation 

Dear Dr. Zhu:

I'm pleased to inform you that your manuscript has been deemed suitable for publication in PLOS ONE. Congratulations! Your manuscript is now with our production department. 

Kind regards, 

on behalf of

Dr. Thippa Reddy Gadekallu 

Academic Editor

PLOS ONE